# Reinforcement Learning for Fail-Operational Systems with Disentangled Dual-Skill Variables

Taewoo Kim [1,2] and Shiho Kim [1,2,*]

1 School of Integrated Technology, Yonsei University, Incheon 21983, Republic of Korea; boratw@yonsei.ac.kr
2 BK21 Program for Intelligent Semiconductor Technology, Yonsei University, Incheon 21983, Republic of Korea
* Correspondence: shiho@yonsei.ac.kr

**Abstract:** We present a novel approach to reinforcement learning (RL) specifically designed for fail-operational systems in critical safety applications. Our technique incorporates disentangled skill variables, significantly enhancing the resilience of conventional RL frameworks against mechanical failures and unforeseen environmental changes. This innovation arises from the imperative need for RL mechanisms to sustain uninterrupted and dependable operations, even in the face of abrupt malfunctions. Our research highlights the system's ability to swiftly adjust and reformulate its strategy in response to sudden disruptions, maintaining operational integrity and ensuring the completion of tasks without compromising safety. The system's capacity for immediate, secure reactions is vital, especially in scenarios where halting operations could escalate risks. We examine the system's adaptability in various mechanical failure scenarios, highlighting its effectiveness in maintaining safety and functionality in unpredictable situations. Our research represents a significant advancement in the safety and performance of RL systems, paving the way for their deployment in safety-critical environments.

**Keywords:** reinforcement learning; skill-based learning; operational safety; fail-operational system

## 1. Introduction

In artificial intelligence, reinforcement learning (RL) is a critical technique that allows agents to learn and improve their actions through interaction with their environment [1]. This method is increasingly vital in various robotic areas and crucial for navigating complex real-world challenges [2].

The primary challenge in RL applications is safety assurance. While RL is effective in many cases, it often struggles to foresee or mitigate risks in novel or unpredictable situations [3–5]. In particular, when faced with unanticipated test data, the neural network may exhibit excessive confidence in its predictions [6,7]. In critical tasks like autonomous driving, such overconfidence can lead to severe failures [8].

RL systems have integrated fail-safe mechanisms to address this when handling exceptional circumstances. These systems are designed to switch to a safe state during errors, preventing further damage or risk and ensuring no catastrophic failures under any operating condition [9–11]. This is also mandated by international standards like ISO 26262 [12], requiring fail-safe features for protective responses to functional failures.

However, for dynamic driving tasks in autonomous vehicles or delivery robots, a simple, fail-safe system for functional safety might not be comprehensive for overall driving safety. A fail-safe mechanism typically prevents catastrophic failures by transitioning the

system into a safe state or using a fallback policy, often completely halting operations upon detecting a fault. Sometimes, more sophisticated responses are needed for system failures. For instance, highway safety measures like stopping or waiting for human intervention can be hazardous. Conversely, fail-operational systems ensure continued operation with minimal functionality loss, even during critical failures [11]. A comparative overview of failure management systems is illustrated in Figure 1.

Architectures often rely on structured Fault Detection, Isolation, and Recovery (FDIR) pipelines in conventional fail-operational systems, particularly in safety-critical domains such as aerospace and autonomous driving. Notably, FDIRO [13] extends this paradigm by incorporating an optimization step after recovery to ensure the reconfigured system remains contextually effective. Adapted for autonomous vehicles, this strategy typically employs low-latency failure detection and isolation, followed by system reconfiguration to ensure safety constraints are met. The subsequent optimization phase adjusts component placement (e.g., computation node assignments) based on runtime context to enhance efficiency.

Additionally, hierarchical fallback architectures, such as those presented in [14], propose tiered control and inference strategies. Depending on system confidence and environmental uncertainty, these systems switch between high-performance and low-risk models. Such approaches emphasize system-level redundancy and architectural fallback mechanisms rather than the policy-level adaptability at the core of our approach.

Moreover, redundancy-based techniques, such as analytical redundancy, are commonly adopted in conventional fail-operational systems. For instance, Simonik et al. [15] demonstrate steering angle estimation for automated driving using analytical redundancy to maintain vehicle functionality in the event of sensor failure. These methods ensure fault tolerance using estimation models to replicate failed sensor outputs and maintain control continuity.

In contrast, our contribution introduces an innovative learning and reasoning model that implements a dual-skills approach, distinguishing between 'task skill' and 'actor skill'. The 'task skill' component involves high-level planning, such as determining the vehicle's intended driving path or destination. In contrast, the 'actor skill' governs low-level control actions based on the vehicle's current mechanical state and environment. This separation enables flexible and modular control, where high-level objectives and low-level execution are decoupled, allowing for better accommodating system changes or faults.

The prior work [16] introduced a skill discovery algorithm that learns morphology-aware latent variables without direct compensation. This algorithm adapts to unseen environmental changes, showing zero-shot inference capabilities for morphological alterations. Building on this, we developed a fail-operational system with real-time fault detection and response planning. In case of significant errors due to sensor or system failures, the system updates the actor skill to maintain continuous operation.

The proposed method underscores the significance of skill-based learning in enhancing RL system resilience and reliability. By creating systems that aptly respond to mechanical failures and adapt to changing scenarios, we explore new safety and efficiency avenues in RL for critical systems like autonomous driving, where failure costs are exceptionally high and intolerable.

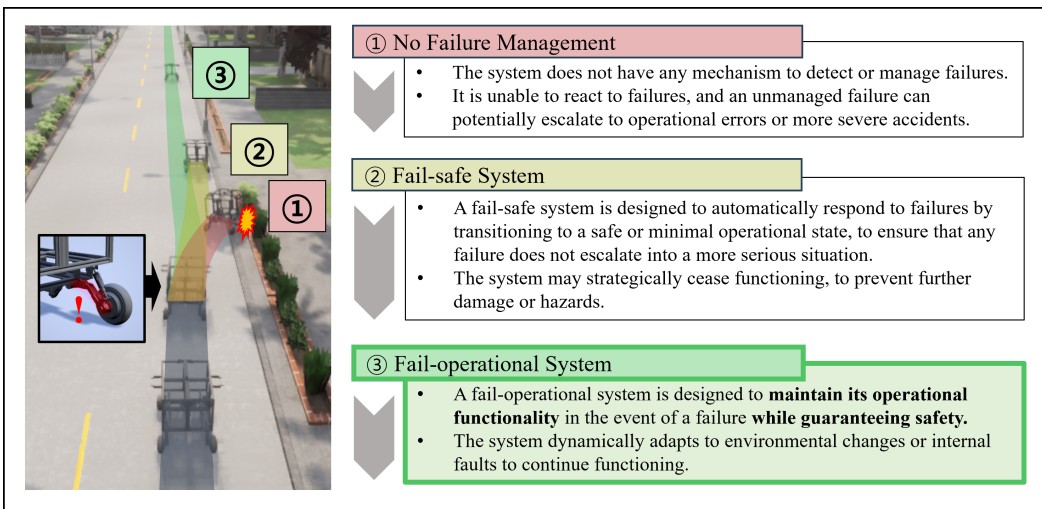

**Figure 1.** Comparative overview of failure management systems.

## 2. Related Works

### 2.1. Safe Reinforcement Learning

Safe Reinforcement Learning (SRL) is a specialized branch of RL that goes beyond the traditional goal of maximizing rewards, including adherence to specific safety constraints within its learning objectives. It emphasizes integrating these constraints into the RL framework, with attention to factors like expected return, risk metrics, and potentially hazardous areas within the Markov Decision Process (MDP) framework [17,18].

A common strategy for addressing SRL challenges is to adjust the policy optimization phase of classic RL algorithms. This adaptation aims to balance the pursuit of task-related rewards with observing safety constraints. Techniques like trust region policy optimization [19,20], Lagrangian relaxation [21–23], and the construction of Lyapunov functions [24,25] have been investigated for this purpose.

SRL also involves limiting the policy's exploration to safer zones within the MDP. This is performed via Recovery RL [17,26,27] or Shielding Mechanisms [28,29], which act as preventive measures against the agent's venture into risky states, thereby maintaining exploration within safe parameters.

Moreover, model-based RL contributes significantly to SRL by facilitating safe exploration and control. It often incorporates Model Predictive Control (MPC) [30], allowing real-time optimization of control tasks with embedded safety constraints, such as state restrictions [31–33] or behavioral shields [29,34].

### 2.2. Fail-Operational Systems

Fail-operational systems are designed to maintain core functionalities even in partial failures—a critical requirement in safety-critical domains such as autonomous driving and aerospace. Traditional fail-operational architectures rely on structured Fault Detection, Isolation, and Recovery (FDIR) pipelines. The extended FDIRO model incorporates an optimization step into the standard FDIR procedure, dynamically reallocating system resources (e.g., computation node assignments) based on contextual information to enhance operational effectiveness after recovery [13].

Hierarchical fallback architectures present another common fail-operational strategy, in which control is tiered according to performance and risk levels. As introduced by Polleti et al. [14], such systems switch between high-performance and conservative models based on system confidence or environmental uncertainty. These approaches offer robustness by relying on predefined control transitions and behavioral backups.

Additionally, analytical redundancy is widely employed to compensate for sensor-level failures. For example, Simonik et al. [15] propose a method for estimating steering angles using redundant models when direct sensor readings are unreliable or unavailable. This approach ensures system continuity by synthesizing missing inputs from other available signals.

While effective, these conventional methods depend heavily on handcrafted redundancy, explicit fallback logic, or architecture-level reconfiguration, which can limit adaptability in highly dynamic or unexpected failure scenarios.

*2.3. Skill-Based Reinforcement Learning*

Skill learning in the context of RL includes a variety of techniques and approaches aimed at improving the policy's adaptability and functionality, allowing the agent to adapt and fine-tune for unseen tasks more efficiently [35,36]. Historical algorithms such as SKILLS [37] and PolicyBlocks [38] pioneered the concept of skill learning. These learned skills are expressed in various forms, such as sub-policies, often referred to as options [39–41], and sub-goal functions [42,43]. More recent developments include embedding skills in continuous space through stochastic latent variable models [44], which concisely represent a wide range of skills [45–47].

Unsupervised skill discovery is an essential aspect of skill learning and is helpful in environments where it is challenging to design explicit rewards or where reward signals are sparse. This approach maximizes the mutual information between skills and states to derive meaningful skills [48,49]. Various algorithms utilize mutual information as an intrinsic reward, optimizing through reinforcement learning [50–52], while other methods learn a transition model and use model predictive control over downstream tasks [53,54].

However, mutual information-based approaches have limitations, especially in covering a broader region in the state space [49]. Recognizing these limitations, several alternative methodologies have been proposed. Lipschitz-constrained Skill Discovery [55] stands out in that it focuses on mapping the skill space to the state space and maximizing the Euclidean travel distance within the state space. Unsupervised goal-conditioned RL learns goals, corresponding tomdiverse reaching policies [56,57], and Controllability-Aware Skill Discovery [49] aims to make agents continuously move to hard-to-control states.

In our proposed method, we introduce the new concept of separated skill discovery and specialize each skill space in a specific domain. This specialization is designed to make skills more targeted and effective in specific areas. It is expected to broaden the scope that each skill can effectively cover and improve its adaptability and applicability.

*2.4. Disentangled Skill Discovery*

In prior research [16], we unveiled an innovative skill discovery algorithm that delineates skill domains to acquire knowledge of morphology-sensitive latent variables without direct adjustments. This algorithm is adept at adjusting to unanticipated morphological changes in the environment, showcasing the ability to perform zero-shot inference for morphological adaptations by seamlessly responding to these variations.

The algorithm's foundation lies in two key elements: the 'task skill' and the 'actor skill'. The 'task skill' pertains to overarching objectives such as charting the vehicle's desired route or destination. It steers the vehicle toward achieving its objectives. Conversely, the 'actor skill' governs precise driving maneuvers based on the vehicle's instantaneous state and its interaction with the environment. The system divides the agent's operational expertise into two separate latent variables, each shaped through two specialized variational autoencoders (VAEs) [44].

Drawing from unsupervised skill discovery methodologies [52–54], notably the maximization of mutual information, the task VAE is designed to create as varied geometric paths as possible. Meanwhile, the action VAE learns the necessary actions for task execution by analyzing state and action pairings, encapsulating this information into an actor latent variable.

The system includes the 'sampler' and the 'explorer', both crucial to learning. The sampler, influenced by information-centric strategies [53,54], generates novel trajectories by maximizing the mutual information between the geometric state space and the latent space. The Explorer's role is to enact the sampler's directives. The training environment employs multiple agents, each with unique physical attributes and Explorer policies custom-fitted to them. This arrangement ensures task learners discern and assimilate the unique behavioral patterns specific to each agent model.

While this algorithm has proven effective in adapting to morphological changes and exhibited zero-shot inference prowess for such modifications, we recognized a critical shortfall: the urgency of detecting and reacting to malfunctions in real time is paramount for ensuring safety. To address this, this study has led to the creation of an exhaustive fail-operational system. This system is meticulously designed to react immediately to abrupt environmental shifts, preserving the continuity of essential functions even amidst component malfunctions.

## 3. Methodology

### 3.1. Skill-Based Fail-Operational System

This study proposes a fail-operational system that can cooperate with RL, focusing on safety and adaptability. We can detect errors and adjust their behavior in real time to maintain stable operation. Figure 2 is a proposed conceptual diagram representing the workflow of the proposed system.

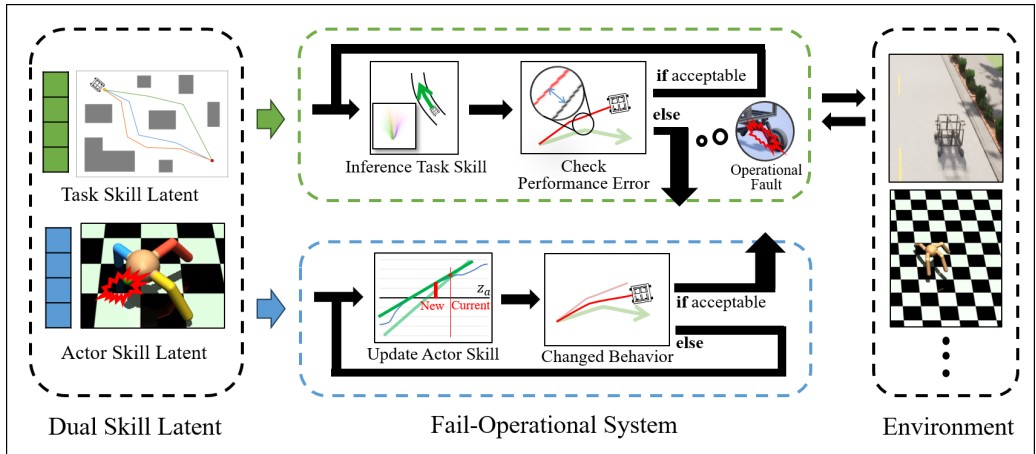

**Figure 2.** Conceptual diagram of the inference strategy of the proposed system.

We propose a fail-operational system that responds appropriately to mechanical problems when they occur and maintains stable driving. This innovative learning and reasoning model implements a dual-skills approach, distinguishing between the 'task skill' and 'actor skill'.

The 'task skill' component is responsible for essential goals such as determining the vehicle's intended driving path or destination. It gives the vehicle the direction it needs to move toward its goal. On the other hand, the 'actor skill' is an element that directs specific driving actions according to the vehicle's current state and surrounding environment. The

actor skill embeds the physical control needed to follow the trajectory generated by the task skill under the current mechanical condition of the system.

During operation, the system monitors the deviations or anomalies from expected performance. If the deviation remains within predefined safety thresholds, the 'actor skill' is maintained, ensuring the vehicle continues driving as expected. However, if a sensor malfunction or drive system defect causes a discrepancy, the system actively responds to the failure. Unlike traditional fallback or redundancy mechanisms, the system updates its 'actor skill' to re-calibrate its driving method to maintain stability without relying on predefined redundant components.

### *3.2. Dual Skill Latents*

This approach involves formulating two distinct disentangled latent variables encapsulating different aspects of the agent's operational knowledge. These variables are essential to the partitioned latent space, which is constructed through the deployment of two dedicated variant autoencoders [44] (task VAE and action VAE).

Following the principles of unsupervised skill discovery [52,54,58], particularly the necessity of maximizing mutual information, the task VAE is able to generate diverse geometric paths that comprehensively represent all feasible trajectories required for operational flexibility. The action VAE uses all state-and-action pairs to learn the explorer's actions to perform the task and stores this information as an Actor latent.

The following sections explain how each learning component is learned. Additionally, hyperparameters and constant values are explained in detail in Appendix A.

### 3.2.1. Task Skill Learner and Sampler

The task VAE generates an array of coordinates representing the agent's movements. Both the input and output of the task VAE consist of an array of geometric locations, and each array spans frame length $T$. The operation system strategically chooses the skill latent $z_t$ to provide the agent's immediate path, enabling decision-making to achieve the task.

The task encoder $q_{\phi_t}(z_t|s_{1:T})$ is designed to identify valid motion trajectories and encode this knowledge within the skill potential $z_t$. At the same time, the task decoder $p_{\theta_t}(\hat{s}_{1:T}|z_t)$ emulates the trajectories discovered in the exploration process. The loss function for the task VAE, denoted as $L_t$, is derived from the Evidence Lower Bound (ELBO) [59], with the prior $r(z_t)$ normalized as a standard Gaussian distribution $N(0,1)$. Additionally, to mitigate potential bias and overfitting during training, we apply KL-divergence regularization within the VAE framework.

$$L_t = -E_{z_t \sim q_{\phi_t}}(\log p_{\theta_t}(s_{1:T}|z_t)) + \beta D_{KL}\big(q_{\phi_t}(z_t|s_{1:T})\|r(z_t)\big) \tag{1}$$

The role of the sampler is to operate according to principles inspired by information-based approaches [51–53]. The sampler $p_{\theta_s}(s_{1:T}|z_t)$ is tasked with generating new trajectories by optimizing the mutual information $I(S; Z_t)$ between the geometrical state space $S$ and latent space $Z_t$. The loss function for sampler parameter $\theta_s$ is designed as follows:

$$L_s = E_{z_t \sim q_{\phi_t}, \hat{s}_{1:T} \sim p_{\theta_t}}[\log p_{\theta_s}(\hat{s}_{1:T}|z_t) - \log p_{\theta_s}(\hat{s}_{1:T})] \tag{2}$$

### 3.2.2. Actor Skill Learner

To encapsulate the actor's state changes within the skill latent space $Z_a$, we employ an action variational autoencoder (VAE), consisting of an action encoder $q_{\phi_a}(z_a|s_{0:T-1}, a_{0:T-1})$ and an action decoder $p_{\theta_a}(\hat{a}_{1:T}|s_{0:T-1}, z_t, z_a)$.

The encoder processes the trajectories of actions and states generated through the execution of the explorer, extracting these experiences into the skill latent $z_a$. Conversely,

the action decoder requires access to both $z_t$ and $z_a$ to reconstruct the actor's driving maneuver. In particular, the construction of the decoder differs from the standard VAE as it incorporates both $z_a$ and $z_t$ as inputs. The objective function is thus defined as follows:

$$L_a = -E_{z_a \sim q_{\phi_a}, z_t \sim q_{\phi_t}} \left( \log p_{\theta_a}(a_{1:T}|s_{0:T-1}, z_t, z_a) \right) + \beta D_{KL} \left( q_{\phi_a}(z_a|s_{0:T-1}, a_{0:T-1}) \| r(z_a) \right) \quad (3)$$

*3.3. Skill Inferencing*

### 3.3.1. Task Skill

The task VAE generates an array of coordinates representing the agent's movements. Both the input and output of a task VAE consist of an array of geometric locations, and each array spans frame length $T$. The operation system strategically chooses the skill latent $z_t$ to provide the agent's immediate path, enabling decision-making to achieve the task.

The $z_t$ space contains a diverse range of potential paths originating from the agent's current location, encompassing all possible trajectories that the agent may undertake in the future. At every moment, the system continuously infers the optimal $z_t$ corresponding to the path that most closely aligns with a given target path or point. The mathematical definition of optimal $z_t$ is as follows. $G$ is the provided or calculated target path, and $d(\cdot, \cdot)$ is the distance between the path and the point.

$$\underset{z_t}{\operatorname{argmin}} \sum_{i=1}^{T} d(G, \hat{s}_i^{loc}), \quad where \quad \hat{s}_{1:T}^{loc} \sim p_{\theta_t}(\cdot|z_t) \quad (4)$$

To determine the most suitable $z_t$, the system samples $z_t$ in the latent space and evaluates its alignment with the desired target path. We utilized a precomputed table of pairs of specific instances of $z_t$ and their corresponding paths. The table serves as a lookup mechanism that can quickly identify the optimal $z_t$. The feasibility of this approach is grounded in the fact that the task skill is independent of the actor's attributes, which we may wish to alter during operation. This independence ensures that the table remains valid regardless of changes in the actor's state or configuration.

### 3.3.2. Actor Skill

The actor skill is designed to improve autonomous agents' operational capabilities and safety. It reacts to environmental interactions in the context of an agent's mechanical properties and their impact on performance.

The system continuously monitors the actual path of the agent and compares it with the intended path. Mathematically, the performance error $\varepsilon$ is defined as follows:

$$\varepsilon = \sum_{i=1}^{T} d(\hat{s}_i^{loc}, s_i^{loc}) \quad where \quad \hat{s}_{1:T}^{loc} \sim p_{\theta_t}(\cdot|z_t) \quad (5)$$

$d(\cdot, \cdot)$ is the Euclidean distance between two points. Note that $\hat{s}$ is the output of the task decoder, and $s$ is the observation value after the system performs the action by the action decoder.

In the absence of mechanical errors, an agent's existing actor skill latent $z_a$ should be preserved. It represents the agent's optimal response method in the current state and is initialized through a latent sampling process. The initial value of $z_a$ is precomputed as follows:

$$z_a = E_{z_t} \left\{ \sum_{i=1}^{T} d(\hat{s}_i^{loc}, s_i^{loc}) \right\} = E_{z_t}(\varepsilon) \quad (6)$$

Whether to maintain or update the actor skill is determined by predefined safety thresholds $\varepsilon_{th}$. If the observed error $\varepsilon$ in the executed skill is within the threshold, the current actor skill is still considered acceptable and maintained.

However, if the deviation exceeds a safety threshold, the system initiates an update process for actor skills. To infer the actor skill $z_a$, we use a technical approach that relies on dynamic adjustment of $z_a$ in response to observed performance errors $\varepsilon$. This process utilizes the scaling factor $\alpha$. The first time, $\alpha$ is initialized to the initial value $\alpha_{init}$ based on experience.

Firstly, $z_a$ is updated by adding the product of the performance error $\varepsilon$ and $\alpha$. The parameter $\alpha$ is then modified based on the Exponential Moving Average of $\Delta z_a / \Delta \varepsilon$. The adjustment fine-tunes $\alpha$ to ensure that it accurately reflects the relationship between actor skill changes and performance error changes. Mathematically, this is represented as follows:

$$z_a \leftarrow z_a + \alpha \cdot d \tag{7}$$

$$\alpha \leftarrow \alpha(1 - \tau) + \frac{\Delta z_a}{\Delta \varepsilon} \tau \beta \tag{8}$$

where $\tau$ is the smoothing factor. And since $d$ is an actual observation value and has a lot of noise, $\beta$ is a hyperparameter to achieve stability by reducing the reactivity of $z_a$. To improve the robustness of this process, especially when there is a risk of local minima during the search, we introduce a strategy to set several pivotal points as search criteria based on previously learned actor settings. The search range is extended to other pivots after searching around the initial $z_a$ if the performance error does not fall below a threshold. The search process is mathematically structured as follows.

Initially, determine the set of pivot points $P = \{p_1, p_2, \cdots, p_n\}$ for $z_a$. This can be inferred from the task performance results of a pre-learned actor or set to divide the $z_a$ space uniformly. If $\alpha$ becomes too large, it will no longer be possible to find a point with a low error value even if $z_a$ is updated (see Figure 3). To determine this, we set a threshold $\alpha_t h$ that the alpha value can have. If $\alpha$ becomes greater than $\alpha_{th}$ even though the error is less than the $\varepsilon_{th}$, $z_a$ jumps to one of the nearby pivots. The entire process of inferring skill is described in Algorithm 1.

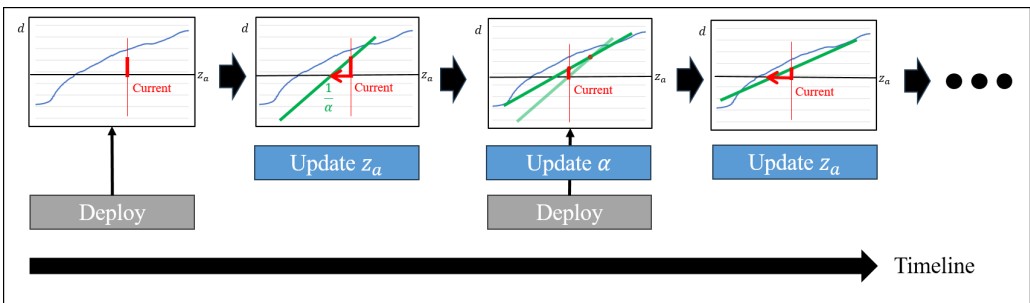

**Figure 3.** A diagram showing the process of updating actor skills. $z_a$ is updated based on observed error and the parameter $\alpha$ is adjusted with the change in error during the update process. If $\alpha$ becomes larger than $\alpha_t h$, $z_a$ jumps to one of the nearby pivots.

---

**Algorithm 1** Skill inference algorithm

---

1:  Load Parameters $\theta_t, \theta_a$
2:  Initialize $z_t, \alpha, P = \{p_1, p_2, \cdots, p_n\}$
3:  Precompute $p_{\theta_t}(\hat{s}^{loc}|z_t)$ for $z_t \sim [-2,2]^{\dim z_t}$
4:  **while** not finished **do**
5:      Optain $G$ from environment
6:      Find $z_t$ from Equation (4)
7:      Deploy action to the environment
8:      Optain $\varepsilon$ from environment and Equation (5)
9:      **if** $z_a$ was updated in the previous step **then**
10:          Update $z_a$ using Equation (8)
11:      **end if**
12:      **if** $\varepsilon > \varepsilon_{th}$ **then**
13:          **if** $\alpha < \alpha_{th}$ **then**
14:              Update $z_a$ using Equation (7)
15:          **else**
16:              Update $z_a$ to nearest $p_i \in P$
17:          **end if**
18:      **end if**
19:  **end while**

---

## 4. Experiments

### 4.1. Skill Discovery from Predefined Models

In the experiment, we evaluated the effectiveness of the proposed algorithm in adapting to unseen changes in mechanical states through experiments conducted within the CARLA vehicle simulator [60]. For this experiment, we utilized a four-wheeled delivery robot model. To enable the system to respond to a variety of mechanical conditions, we designed a standard model $m_1$, a model with an enlarged left wheel $m_2$, and a model with an enlarged right wheel $m_3$ (see Figure 4). These variations are intended to simulate problems due to asymmetric wheel configurations.

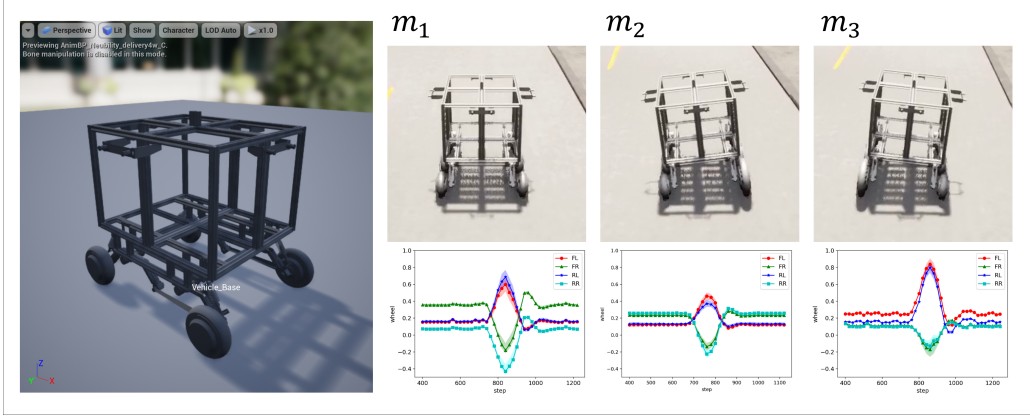

**Figure 4.** (**Left**) The design of the four-wheeled robot used in the experiment. (**Top Right**) Graphical representation of the shapes of three predefined robot models. (**Bottom Right**) Optimal input values for each wheel of the corresponding model to make an ideal left turn.

For additional details and to access the source code and results of the experiments, visit our repository https://github.com/boratw/sd4m_carla (accessed on 8 March 2025).

### 4.2. Visualization of Skill Latent Space

#### 4.2.1. Task Skill Latent

The main learning goal for the task latent variable $z_t$ was to enable the system to learn all possible paths a vehicle can take autonomously. Figure 5 shows the visualization of

embedded task skills. After training, $z_t$ becomes a repository of potential routes containing the different navigation choices available to the vehicle based on its current location. It was confirmed that all possible paths in the forward direction of the robot were well embedded. In the experimental configuration, the trajectory length $T$ was set to 500 ms.

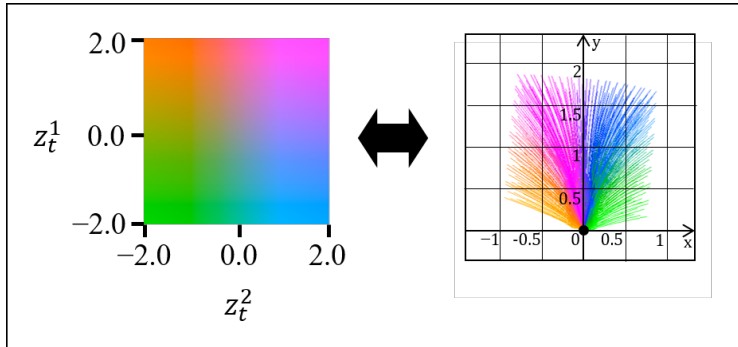

**Figure 5.** Visualization of embedded task skills. The left side of the figure shows the relationship between latent value and color, and the right side demonstrates the corresponding target paths decoded from the latent.

### 4.2.2. Actor Skill Latent

In contrast, the learning objective for the actor latent variable $z_a$ focuses on the vehicle's shape. Specifically, the goal is to ensure that each distinct physical characteristic of the vehicle is appropriately mapped to an optimal distance within the latent space. This mapping ensures that changes in the vehicle's physical characteristics, such as different wheel sizes or asymmetric configurations, are accurately represented and distinguishable within $z_a$.

Figure 6 and Table 1 show the performance of the proposed algorithm and the SAC [61] agent for each predefined shape. $z_a$ is the optimal $z_a$ value obtained through Equation (7), and the velocity and error values are the average values shown in the lane keeping scenario, which will be described later.

**Table 1.** Comparison of performance between the proposed algorithm and SAC agent for predefined shapes.

| Shape | Algorithm | $z_a$ | Velocity (m/s) | Error (cm) |
|:-----:|:---------:|:-----:|:--------------:|:----------:|
| $m_1$ | Ours | 0.37 | 3.22 | 2.52 |
|       | SAC  | N/A  | 2.41 | 0.47 |
| $m_2$ | Ours | −0.18 | 3.57 | 1.94 |
|       | SAC  | N/A   | 2.28 | 1.32 |
| $m_3$ | Ours | 1.01 | 2.69 | 1.05 |
|       | SAC  | N/A  | 2.46 | 0.84 |

We confirmed that the learner we trained learned the capability of operating at a similar level to the SAC policy, which focused on learning only each shape. And $z_a$ values were also mapped separately from each other, as expected.

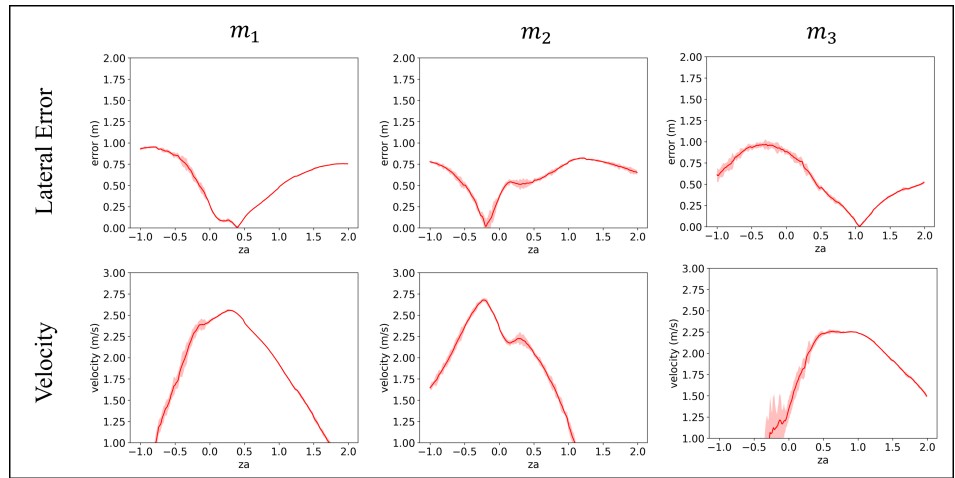

**Figure 6.** Relationship between the actor latent variable $z_a$ and task performance metrics across the shape of the vehicle.

### 4.3. Evaluation of Robustness in Static Environment Changes

To evaluate the adaptability of the learned model in response to failure scenarios, we designed experiments focusing on a four-legged robot under various compromised conditions. The goal was to evaluate the system's ability when specific mechanical failures occur and compare its performance to existing reinforcement learning models.

We prepared four separate models of a four-wheeled robot (Figure 4), each representing a unique failure state. In these models, each corresponding robot wheel was individually disconnected from the linked motor, which means that the input value is not sent to the wheel.

For each model, we determined the optimal $z_a$ value, as described in Equation (7). The identified $z_a$ that best compensated for the specific model ensured that the robot could effectively perform its intended task despite the damaged wheels.

The driving capabilities of the system juxtaposed with those of standard reinforcement learning models were evaluated across four driving scenarios: Lane Keeping, Left Turn, Right Turn, and Lane Change. The primary metric for evaluation is lateral distance, measured as the deviation between the robot's actual position and the optimal path provided to the system. Figure 7 shows a representative example of experiments focusing on a model with a disconnected "Front Left" wheel. And Figure 8 and Table 2 present a detailed comparison of the results for each scenario across the different models.

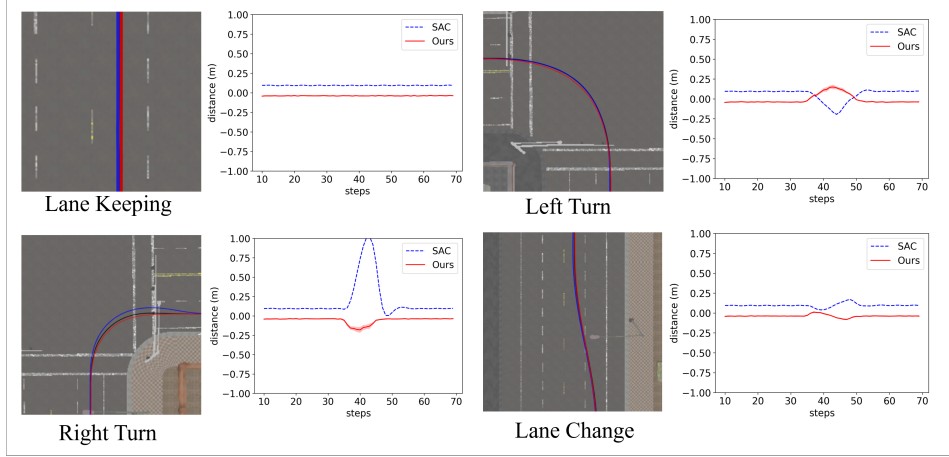

**Figure 7.** Visualizations of the driving scenarios that compare the paths driven by the SAC model (blue line) and our algorithm (red line), using a model with a broken 'Front Left' wheel. The black line represents the ideal trajectory generated by the Task VAE. The right side of each visualization displays the observed lateral distance values during the experiment.

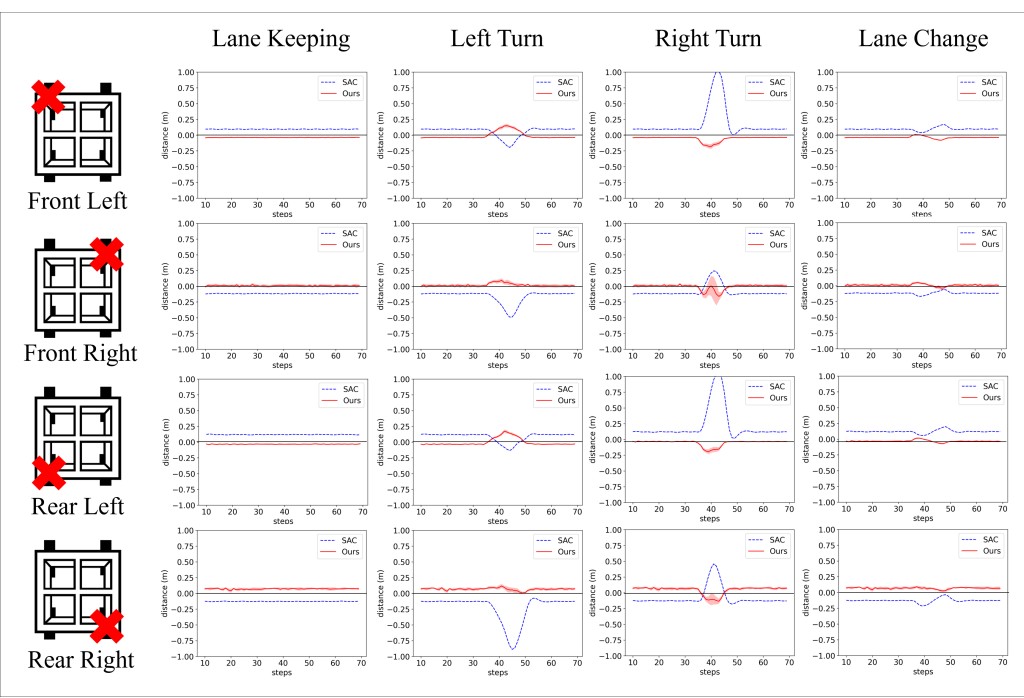

**Figure 8.** The lateral deviation from the reference trajectory for each model. The data were averaged across different scenarios based on 25 experiments, and the shaded area in the graph indicates the 95% confidence interval for the value.

**Table 2.** Comparative analysis of performance between the proposed algorithm and the SAC with the optimal $z_a$ for each kinematic modification of wheel size.

| Disconn. Wheel | Algorithm | | Lane Keeping Error (cm) | | Vel (m/s) | Turn Left Error (cm) | | Vel (m/s) | Turn Right Error (cm) | | Vel (m/s) | Lane Change (Left) Error (cm) | | Vel (m/s) |
|---|---|---|---|---|---|---|---|---|---|---|---|---|---|---|
| | Name | $z_a$ | Avg | Max | Avg | Avg | Max | Avg | Avg | Max | Avg | Avg | Max | Avg |
| Front Left | Ours | 1.19 | 3.8 | 4.16 | 2.5 | 5.71 | 14.95 | 2.51 | 5.71 | 18.06 | 2.41 | 4.1 | 8.11 | 2.5 |
| | SAC | N/A | 9.43 | 9.85 | 2.34 | 9.32 | 19.41 | 2.33 | 23.97 | 102.1 | 2.25 | 9.92 | 17.09 | 2.33 |
| Front Right | Ours | −0.37 | 0.97 | 1.67 | 2.45 | 2.01 | 9.25 | 2.38 | 2.33 | 16.11 | 2.47 | 1.29 | 4.25 | 2.45 |
| | SAC | N/A | 11.75 | 12.18 | 3.47 | 18.98 | 49.02 | 3.32 | 12.62 | 24.44 | 3.36 | 11.51 | 16.79 | 3.46 |
| Rear Left | Ours | 1.13 | 3.26 | 3.69 | 2.49 | 5.52 | 17.39 | 2.49 | 5.5 | 19.35 | 2.39 | 3.53 | 7.33 | 2.48 |
| | SAC | N/A | 12.04 | 12.54 | 2.39 | 9.96 | 13.42 | 2.37 | 27.07 | 105.4 | 2.3 | 12.28 | 19.88 | 2.37 |
| Rear Right | Ours | −0.73 | 7.23 | 7.88 | 2.44 | 6.54 | 11.72 | 2.25 | 7.41 | 13.42 | 2.48 | 6.6 | 8.17 | 2.44 |
| | SAC | N/A | 12.69 | 12.98 | 3.18 | 28.54 | 88.79 | 3.03 | 16.22 | 45.79 | 3.11 | 12.53 | 21.35 | 3.19 |

Since no scenario in which the robot's wheels were disconnected was presented during the training phase, the system inferred the optimal $z_a$ value even though it was a completely new failure situation. Additionally, utilizing the inferred $z_a$ values, the system demonstrated resilience by performing at a similar level to a scenario where the robot's wheels were intact.

### 4.4. Evaluation of Fail-Operational System

Building on the previous experiment, which evaluated the adaptability of models under static failure scenarios, we extended the investigation to include more dynamic and unpredictable conditions. Specifically, we assessed the system's responsiveness to unexpected mechanical failures during operation.

First, we examined the vehicle's reaction to a sudden change in wheel size. The left and right wheels were altered simultaneously, requiring different inputs to maintain a straight trajectory. Initially, the delivery robot was driven straight, with all wheels at 100% of their original size. Subsequently, the size of one wheel was varied by a certain percentage.

We measured the maximum deviation of both the Soft Actor–Critic (SAC) algorithm and the proposed algorithm from the center of the lane, as well as the time required to return to parallel driving. The results of this experiment are presented in Figure 9 and Tables 3 and 4. As indicated in the tables, the proposed method demonstrates significantly lower lateral deviation and faster recovery times under various failure scenarios.

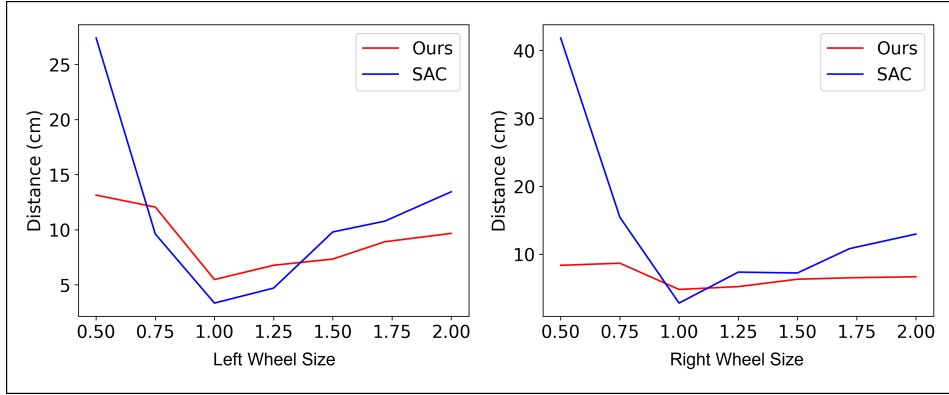

**Figure 9.** Comparative performance of the proposed algorithm and SAC in observing the deviation from a lane's center when the robot is driven straight.

**Table 3.** Analysis of the performance between algorithms in the left wheel size variation experiment.

| Size of Left Wheel | | 50% | 75% | 100% | 125% | 150% | 175% | 200% |
|---|---|---|---|---|---|---|---|---|
| Dist (cm) | SAC | 27.4 | **9.613** | **3.338** | **4.7** | 9.795 | 10.78 | 13.44 |
| | Ours | **13.14** | 12.05 | 5.477 | 6.768 | **7.336** | **8.905** | **9.664** |
| Vel (m/s) | SAC | 1.51 | 2.166 | 2.511 | 2.748 | 2.524 | 2.517 | 2.546 |
| | Ours | 0.792 | 2.502 | 3.007 | 3.067 | 3.181 | 3.187 | 3.249 |
| $z_a$ | Ours | −0.72 | −0.095 | 0.37 | 0.525 | 0.9 | 1.027 | 1.136 |
| Time to recovery (s) | SAC | X | 1.25 | - | 0.9 | 0.7 | 1.55 | 1.85 |
| | Ours | 1.05 | 0.9 | - | 0.65 | 0.6 | 0.95 | 1.4 |

Note: Bold entries indicate lower values, which represent better performance in terms of distance error or deviation.

**Table 4.** Analysis of the performance between algorithms in the right wheel size variation experiment.

| Size of Right Wheel | | 50% | 75% | 100% | 125% | 150% | 175% | 200% |
|---|---|---|---|---|---|---|---|---|
| Dist (cm) | SAC | 41.84 | 15.44 | **2.797** | 7.355 | 7.23 | 10.82 | 12.96 |
| | Ours | **8.361** | **8.667** | 4.805 | **5.223** | **6.312** | **6.524** | **6.674** |
| Vel (m/s) | SAC | 1.151 | 2.119 | 2.255 | 2.362 | 2.608 | 2.868 | 2.864 |
| | Ours | 0.632 | 2.538 | 2.962 | 3.077 | 3.181 | 3.189 | 3.244 |
| $z_a$ | Ours | 1.803 | 1.078 | 0.37 | 0.083 | −0.119 | −0.363 | −0.429 |
| Time to recovery (s) | SAC | X | 1.35 | - | 0.5 | 0.85 | 1 | 1.3 |
| | Ours | 1.8 | 0.75 | - | 0.4 | 0.6 | 0.85 | 0.8 |

Note: Bold entries indicate lower values, which represent better performance in terms of distance error or deviation.

Next, we conducted a similar experiment to the one described in Section 4.3, where the delivery robot navigated through predefined scenarios. This time, one wheel suddenly disconnected during each scenario. The experimental setup included four identical scenarios: Lane Keeping, Left Turn, Right Turn, and Lane Change. Initially, all four wheels of the robot functioned normally. During each scenario, one wheel became disconnected unexpectedly. As in previous experiments, the primary metric for evaluation was the lateral

distance between the robot's actual trajectory and the optimal path. Wheel disconnections were triggered at 40 steps into each scenario.

Figure 10 shows a representative example of experiments focusing on a model with a disconnected "Front Left" wheel. This figure shows the trajectory of each algorithm on the map, the lateral errors, and the inferred $z_a$ value. And Figure 11 and Table 5 present a detailed comparison of the results for each scenario across the different models.

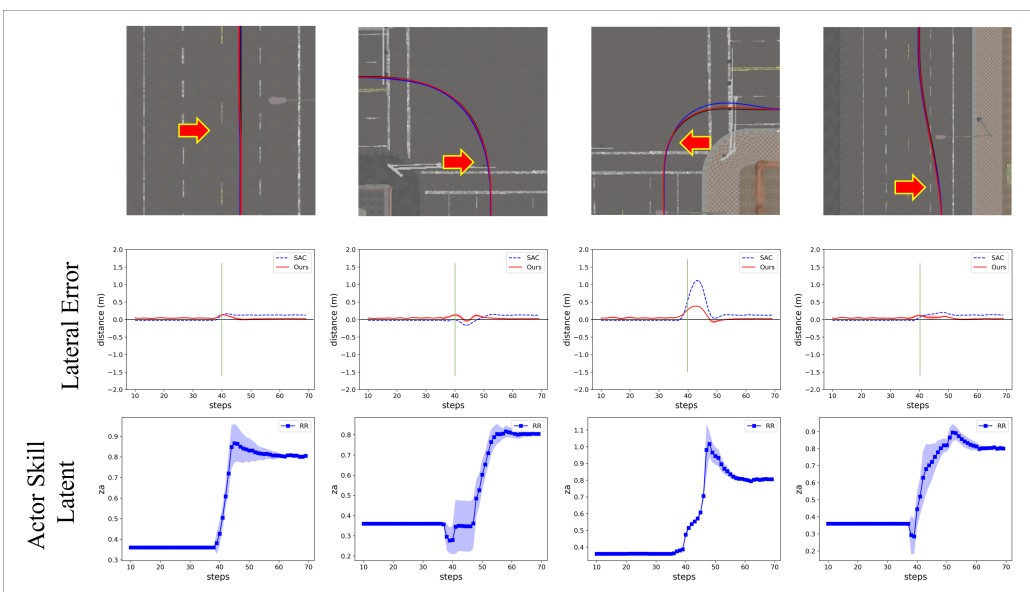

**Figure 10.** The result of the experiment conducted on the 'Front Left' wheel disconnection scenario. (**Top**) Aerial snapshots depict the agent's trajectory before and after the wheel disconnection. The red arrow indicates the point at which the disconnection occurs. (**Middle**) lateral distance for the proposed algorithm versus the SAC model. The green vertical line indicates the step at which the wheel disconnection occurs. (**Bottom**) The inferred $z_a$ from our approach.

Through this experiment, we observed the system's ability to initiate and converge inferences of the agent's skill, $z_a$, when a malfunction occurs. In the "Front Left" case, the $z_a$ value is observed to converge around 0.8, indicating a robust and effective response of the system to mechanical failures. This convergence demonstrates the system's ability to adaptively readjust its strategy in response to sudden changes and maintain optimal performance.

**Table 5.** Comparison of performance between the proposed algorithm and SAC agent.

| Disconn. Wheel | Algorithm | | Lane Keeping | | | Turn Left | | | Turn Right | | | Lane Change (Left) | | |
|---|---|---|---|---|---|---|---|---|---|---|---|---|---|---|
| | | | Error (cm) | | Vel (m/s) | Error (cm) | | Vel (m/s) | Error (cm) | | Vel (m/s) | Error (cm) | | Vel (m/s) |
| | Name | $z_a$ | Avg | Max | Avg | Avg | Max | Avg | Avg | Max | Avg | Avg | Max | Avg |
| Front Left | Ours | 1.19 | 3.81 | 13.51 | 2.74 | 4.93 | 13.41 | 2.83 | 9.05 | 38.54 | 2.56 | 4.51 | 11.74 | 2.7 |
| | SAC | - | 10.29 | 16.65 | 2.5 | 8.6 | 16.8 | 2.5 | 23.4 | 112 | 2.41 | 10.68 | 20.51 | 2.5 |
| Front Right | Ours | −0.37 | 4.19 | 14.1 | 3.38 | 14.32 | 54.18 | 3.13 | 5.67 | 17.88 | 3.25 | 5.31 | 17.89 | 3.36 |
| | SAC | - | 11.01 | 15.58 | 2.46 | 17.82 | 55.95 | 2.37 | 11.49 | 23.94 | 2.47 | 10.96 | 18.9 | 2.46 |
| Rear Left | Ours | 1.13 | 4 | 11.31 | 2.74 | 4.82 | 10.84 | 2.85 | 9.27 | 39.85 | 2.56 | 4.58 | 12.03 | 2.71 |
| | SAC | - | 11.76 | 18.33 | 2.49 | 8.64 | 16.39 | 2.49 | 25.11 | 117.6 | 2.39 | 12.36 | 23.46 | 2.48 |
| Rear Right | Ours | −0.73 | 8.29 | 27.13 | 3.11 | 24.1 | 91.57 | 2.75 | 11.63 | 34.82 | 3.05 | 8.65 | 31.29 | 3.14 |
| | SAC | - | 15.95 | 24.47 | 2.39 | 57.09 | 238.7 | 2.1 | 17.71 | 44.44 | 2.45 | 15.4 | 30.73 | 2.4 |

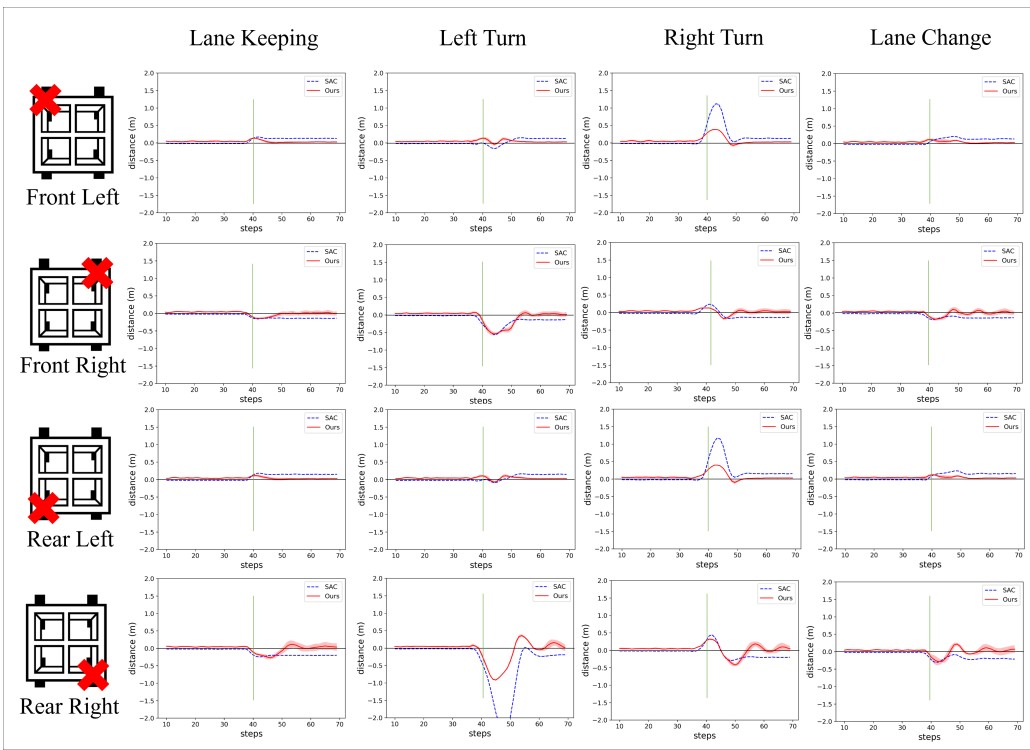

**Figure 11.** The lateral distance for each model across different scenarios. Shading indicates the 95% confidence interval for the value. A green vertical line indicates the step at which the wheel disconnection occurs.

## 5. Discussion

The first experiment was designed around a four-legged robot model subjected to various failure states. The test aims to understand the system's response to specific mechanical failures and benchmark its performance against conventional reinforcement learning models. With this setup, we showed that by using the optimal actor skill variable $z_a$ for each failure state, the robot could continue its task effectively despite unexpected sudden damage to the wheel motor.

In the second experiment, the robot started with all wheels operational, but one wheel would disconnect while running various driving scenarios. This unpredictability tested the system's ability to adjust and readjust its strategy rapidly. The system demonstrated a robust ability to quickly infer and adapt the skill variable $z_a$ in response to these sudden faults in the actuating system. The proposed system could continue performing its task without compromising safety.

A critical insight from the experiments is the system's ability to complete tasks safely during unexpected mechanical failures. This finding highlights the importance of fault-operating systems when human intervention is impractical, but immediate response is essential. For example, in the second experiment, a sudden malfunction at an intersection may lead to significant traffic disruptions and dangers. In such cases, the capacity to continue safe operations takes precedence over maintaining high performance levels.

Although the proposed system demonstrates robustness to mechanical failures, it does not directly address cases involving advanced sensor degradation or severe observation noise. The current framework assumes a minimally sufficient level of perceptual input and focuses on motor-level adaptability. Incorporating observation uncertainty modeling, sensor fusion with redundancy, and robust representation learning could enable the system to reason under imperfect observations. These extensions would further enhance the system's applicability in real-world safety-critical domains.

## 6. Conclusions

In this study, we developed a separable skill-learning architecture and a comprehensive system for real-time fault situation inference and response planning. This study aimed to address the limitations of existing RL methods in predicting and preventing risks in unseen or unpredictable situations in real-world applications.

We evaluated the proposed system in various scenarios involving mechanical faults. The system demonstrated robustness in the experiments by rapidly adapting its behavioral strategy in response to sudden mechanical faults during operation. We showed the system's ability to maintain task performance through improved lateral stability and faster recovery under unexpected fault conditions. This highlights its effectiveness in scenarios demanding an immediate and safe response, where the ability to continue operation safely is critical.

In conclusion, this study demonstrates a novel approach to enhancing the safety and adaptability of RL systems. The ability to handle unexpected scenarios and maintain operational integrity represents a significant advance toward deploying RL agents in real-world environments where unpredictability and safety issues are prevalent. The proposed method is suitable for real-world deployments, including medical robots and autonomous vehicles, where unpredictability and safety issues are prevalent. This study represents a significant step forward in the practical application of RL agents in safety-critical environments.

**Author Contributions:** Conceptualization, S.K.; funding acquisition, S.K.; methodology, T.K. and S.K.; project administration, S.K.; resources, S.K.; supervision, S.K.; validation, T.K.; visualization, T.K.; writing—original draft, T.K.; writing—review and editing, S.K. All authors have read and agreed to the published version of the manuscript.

**Funding:** This research was funded by the Institute for Information and Communications Technology Promotion grant number 2022-0-00966.

**Institutional Review Board Statement:** Not applicable.

**Informed Consent Statement:** Not applicable.

**Data Availability Statement:** The original data presented in this study are openly available in Github at https://github.com/boratw/sd4m_carla (accessed on 8 March 2025).

**Acknowledgments:** This work was supported by Institute of Information & communications Technology Planning & Evaluation (IITP) grant funded by the Korean government(MSIT) (No.2022-0-00966, Development of AI processor with a Deep Reinforcement Learning Accelerator adaptable to Dynamic Environment).

**Conflicts of Interest:** The funders had no role in the design of the study; in the collection, analyses, or interpretation of data; in the writing of the manuscript; or in the decision to publish the results.

## Appendix A. Technical Detail

*Appendix A.1. Environment Setting*

We trained and validated the proposed system within the CARLA [60] simulation environment. To pursue a more detailed control approach suitable for our research purposes, we designed a custom vehicle controller for the simulation. This controller bypasses CARLA's default vehicle control mechanisms, which typically rely on a steering wheel and engine model. Instead, it directly controls the motor of each individual wheel.

During the training and evaluation process, the CARLA simulator provides the state vector of each vehicle, and the system calculates and outputs the optimal action for each state encountered.

- State Vector: The state vector consists of a total of 10 elements. All values are given as coordinates relative to the geometric state at the starting point of the trajectory.

  – Position (x, y): the relative coordinates of the vehicle's current position (m)
  – Speed (x, y): instantaneous speed along the x and y axes (m/s).
  – Acceleration (x, y): acceleration components of the vehicle (m/s$^2$).
  – Vehicle orientation (unit vector x, y): this unit vector represents the orientation of the vehicle in a two-dimensional plane.
  – Vehicle pose (roll and pitch): these quantify the angular orientation of the vehicle in terms of roll and pitch (degree).

- Action Vector: The action vector consists of four unique values ranging from −1 to 1, each value corresponding to a control input provided to each wheel of the vehicle. Positive values indicate acceleration, and negative values represent deceleration.

Interactions between the environment and the system occur every 1/20th of a second. This interval constitutes a single time unit within the experimental setup.

*Appendix A.2. Implementation Details*

The main values and hyperparameters of the learning network are as follows. The architectural framework and hyperparameters used in the learning network are derived with reference to the established structure described in SD4M [16]. As a result, the mathematical expressions of our network are broadly consistent with those described in SD4M [16].

**Table A1.** Parameters of the task and action VAE.

| Parameter | Value |
| --- | --- |
| Encoder [1] | [256, 256] |
| Decoder [1] | [256, 256] |
| $c_t$ [2] | 0.001 |
| Learning rate | 0.0001 |
| $\alpha_a$ [2] | 0.02 |
| $\alpha_d$ [2] | 0.01 |
| $\tau$ [3] | 0.25 |
| $\beta$ [3] | 0.9 |

[1] The number of neurons present in each layer of the MLP. [2] This constant is introduced in [16]. [3] This constant is described in Section 3.3.

**Table A2.** Parameters of the sampler.

| Parameter | Value |
| --- | --- |
| Decoder [1] | [256, 256] |
| $c_t$ [2] | 0.001 |
| Learning rate | 0.0001 |
| $\alpha_a$ [2] | 0.02 |
| $\alpha_d$ [2] | 0.01 |

[1] The number of neurons present in each layer of the MLP; [2] this constant is introduced in [16].

**Table A3.** Parameters of the explorer.

| Parameter | Value |
| --- | --- |
| Value Network [1] | [256, 256] |
| Policy Network [1] | [256, 256] |
| $\gamma$ [2] | 0.95 |
| Initial Entropy [2] | 0.1 |
| Target Entropy [2] | $e^{-4}$ |
| Learning rate | 0.0001 |
| Policy Update rate [2] | 0.05 |

[1] The number of neurons present in each layer of the MLP; [2] this constant is introduced in SAC [61].

## Appendix B. Experiment Settings

For the experiment, we prepared a delivery robot model. This model has four wheels, but unlike a regular vehicle, each can be controlled separately. To achieve this, we modified the power transmission structure inside the collar to send inputs to each wheel individually. All training and experimental code can be found at https://github.com/boratw/sd4m_carla (accessed on 8 March 2025)

### Appendix B.1. Training

The training was conducted by running 25 vehicles simultaneously in an empty lot, collecting results, and learning from them. The training was performed without any reward, utilizing three models: $m_1$, $m_2$, and $m_3$. For training, we ran 1000 experiments, each consisting of 2000 frames, to collect a total of 2 million frames. These frames were divided into 50-frame increments and used as single batches of paths. The network was trained for 350 epochs, each consisting of 64 paths per batch and 32 batches.

### Appendix B.2. Scenario Experiments

This section details the scenario experiments described in Sections 4.3 and 4.4. Each experiment was conducted at a single intersection in the CARLA built-in map Town05. The results were derived from the mean and standard deviation after 25 repetitions. In each experiment, a delivery robot followed 80 waypoints through an intersection, with a scheduled disconnection at the 40th waypoint.

Initially, $z_a$ was set to 0.37, a default value obtained from a previous experiment. The alpha parameter was set to 1.5, and the pivots were $-0.18$, 0.37, and 1.01. However, the pivots were not used as they did not fall into a local minimum during the experiment. The error value was calculated as the vertical distance between an imaginary line connecting the waypoints and the vehicle's position. In contrast, the velocity was calculated as the average value until the end of the episode.

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
