# Peer review of "Reinforcement Learning for Fail-Operational Systems with Disentangled Dual-Skill Variables"

_technologies, doi:10.3390/technologies13040156_

Round 1

Reviewer 1 Report

Comments and Suggestions for Authors

In this paper, the authors present a novel reinforcement learning (RL) framework for fail-operational systems by introducing disentangled dual-skill variables. Their approach aims to enhance the resilience of RL models in safety-critical applications by separating task skill and actor skill, allowing the system to adapt to mechanical failures while maintaining operational integrity dynamically. To further refine the work and enhance its clarity and impact, consider the following suggestions:

  1. The title mentions "Disentangled Dual Skill Variables.” How do these variables fundamentally differ from conventional skill representations in reinforcement learning (RL)?
  2. The authors in the abstract claim that the proposed method "significantly boosts resilience"—what specific quantitative metrics support this claim compared to baseline RL models?
  3. The abstract mentions that the system ensures "uninterrupted, dependable operations." How does it balance fault tolerance and performance degradation when failures occur?
  4. The authors highlight that RL systems often struggle with safety assurance. What are the main limitations of existing fail-safe RL approaches that your work addresses?
  5. How do the authors propose a system that handles sensor failures differently from conventional fallback or redundancy-based safety mechanisms?
  6. The authors introduce a dual-skill approach with task skill and actor skill. How do these two components interact dynamically when encountering unforeseen mechanical failures?
  7. The methodology section describes two Variational Autoencoders (VAEs) for encoding task and actor skills—how do you ensure that these VAEs do not introduce unnecessary biases or overfitting?
  8. The Task Skill Learner generates geometric paths using VAEs. How does the system differentiate between valid trajectory variations and errors caused by model uncertainty?
  9. The manuscript proposes a method that incorporates mutual information maximization. Can you provide an intuition for how this improves skill learning and failure recovery in real-time applications?
  10. What is the computational overhead of continuously updating the actor skill latent variable (za) in real-time, and how does it compare to traditional model-based RL approaches?
  11. In Figure 3, you illustrate the process of updating actor skills based on α and pivots; how sensitive is this adaptation process to hyperparameter tuning?

Comments on the Quality of English Language

The paper is well-written, but some grammatical inconsistencies and awkward phrasing should be refined for clarity. Proofreading and improving sentence structure would enhance readability.

Author Response

Point-by-point response to Comments and Suggestions for Authors

Comment 1: The title mentions "Disentangled Dual Skill Variables." How do these variables fundamentally differ from conventional skill representations in reinforcement learning (RL)? 
Response 1: Thank you for this valuable comment. Our approach differs fundamentally from conventional methods by clearly separating high-level route planning ("task skill") and low-level actuator control ("actor skill") into two disentangled skill variables. This separation allows the system to respond better to unexpected failures. 

Comment 2: The authors in the abstract claim that the proposed method "significantly boosts resilience"—what specific quantitative metrics support this claim compared to baseline RL models?
Response 2: Thank you for bringing this to our attention. The quantitative evidence supporting resilience improvements appears in Tables 2, 3, and 4. Our method consistently shows lower lateral errors and faster recovery during mechanical failures than the baseline SAC model. We have highlighted these quantitative results in the Experiments section. (page 12, line 363)

Comment 3: The abstract mentions that the system ensures "uninterrupted, dependable operations." How does it balance fault tolerance and performance degradation when failures occur?
Response 3: Thank you for this insightful comment. Our approach prioritizes fault tolerance, ensuring safety and operational continuity in the event of failures. We observed only minor performance degradation, such as slightly lower velocity, which demonstrates the effectiveness of our method in striking a balance between high fault tolerance and acceptable performance levels.

Comment 4: The authors highlight that RL systems often struggle with safety assurance. What are the main limitations of existing fail-safe RL approaches that your work addresses?
Response 4: We appreciate this question. Traditional fail-safe RL typically halts operation during faults, limiting continuous operation. Our approach dynamically adjusts skills to allow the system to operate safely despite unexpected failures. We addressed this clearly in the Introduction Section (page 2, line 34)

Comment 5: How do the authors propose a system that handles sensor failures differently from conventional fallback or redundancy-based safety mechanisms?
Response 5: Thank you for bringing this to our attention. Our method dynamically adjusts learned skills in real-time to respond to sensor failures, differing from conventional redundancy or predefined fallback methods. Whereas traditional fail-operational systems emphasize reactive architectural robustness, our method enables intelligent behavioral control through the modular decomposition of skills.  We have expanded the discussion in the Introduction and Related Work sections to include detailed comparisons with widely used fail-operational system designs. (page 2, line 40 and page 3, line 98)

Comment 6: The authors introduce a dual-skill approach with task skill and actor skill. How do these two components interact dynamically when encountering unforeseen mechanical failures?
Response 6: Thank you for highlighting this. Briefly summarize the interaction between the two components: the "task skill" remains stable, defining the intended path. When a mechanical failure occurs, the "actor skill" latent variable dynamically adjusts in response to the deviation from the intended path. Specifically, the system measures the performance error (ε) and updates the actor's skill using a linear adjustment (equation (7)), ensuring that the actual path remains close to the ideal trajectory from the task skill.

Comment 7: The methodology section describes two Variational Autoencoders (VAEs) for encoding task and actor skills—how do you ensure that these VAEs do not introduce unnecessary biases or overfitting?
Response 7: Thank you for highlighting this concern. Our manuscript utilized standard VAE training techniques, including KL-divergence regularization, to avoid biases and overfitting, as recommended in prior literature on VAE implementations [41]. Additionally, a similar methodology for disentangled skill discovery was successfully demonstrated in prior research [17], ensuring the reliability of our training method. We stated this point clearly in the Methodology section (page 6, line 221)

Comment 8: The Task Skill Learner generates geometric paths using VAEs. How does the system differentiate between valid trajectory variations and errors caused by model uncertainty?
Response 8: Thank you very much for your question. We want to clarify that, in our system, the Task Skill Learner generates an ideal reference trajectory. Under normal circumstances, the Actor's Skill precisely follows this trajectory. Therefore, any deviation from the trajectory the task learner produces is considered a fault condition.

Comment 9: The manuscript proposes a method that incorporates mutual information maximization. Can you provide an intuition for how this improves skill learning and failure recovery in real-time applications?
Response 9: Thank you for this insightful comment. Without mutual information maximization, the Task VAE would fail to represent all feasible trajectories required for operational flexibility. Therefore, mutual information maximization is a fundamental component of our methodology. We added this explanation to the manuscript. (page 6, line 205)

Comment 10: What is the computational overhead of continuously updating the actor skill latent variable (za) in real-time, and how does it compare to traditional model-based RL approaches?
Response 10: We appreciate your practical consideration. The computational overhead associated with updating our method's actor skill latent variable (za) is minimal because it involves only a straightforward linear equation. This simplicity naturally implies low computational costs, making it suitable for real-time applications. 

Comment 11: In Figure 3, you illustrate the process of updating actor skills based on α and pivots; how sensitive is this adaptation process to hyperparameter tuning?
Response 11: Thank you for your valuable observation. The critical hyperparameter affecting this adaptation process is β, and we selected β as listed in Table A1, specifically β = 0.9. The choice of a large β enables the system to respond quickly to unexpected mechanical faults, although this may slightly reduce the precision of the resulting actor skill adjustments. Given our primary goal of promptly addressing mechanical failures, we intentionally selected this β value.

Response to Comments on the Quality of English Language

Point 1: The paper is well-written, but some grammatical inconsistencies and awkward phrasing should be refined for clarity. Proofreading and improving sentence structure would enhance readability.
Response point 1: We sincerely appreciate the reviewer's feedback on the overall writing quality. 
We acknowledge that certain sections may contain grammatical inconsistencies or awkward phrasing that could affect clarity. In response, we have conducted a thorough proofreading and editing pass to refine the sentence structure, improve fluency, and enhance readability throughout the manuscript. We will ensure that the revised version presents the technical content more clearly and maintains a professional tone consistent with publication standards. Thank you for pointing this out.

Reviewer 2 Report

Comments and Suggestions for Authors

This manuscript presents a novel reinforcement learning (RL) approach designed for fail-operational systems in safety-critical applications. The core idea is to disentangle dual skill variables to enhance the adaptability of RL systems when facing mechanical failures and unexpected environmental changes. 

The proposed method demonstrates innovation and practical significance; however, the following issues need to be addressed:

1.This manuscript inconsistently uses terms such as "this study", "our work", and "our research". It is recommended to maintain consistency in terminology throughout this manuscript to improve academic clarity and readability.
2.Lines 139–147 introduce task skill and actor skill, but their interrelationship is not clearly explained. It is suggested to elaborate on their connection and provide a more detailed explanation of Figure 2 to enhance clarity.
3.This manuscript mentions that the proposed method is applicable to safety-critical systems such as autonomous driving but does not provide specific case studies or application scenarios. It is recommended to expand the discussion in the Conclusion section to highlight the practical implications of the method.
4.Absence of quantitative experimental evidence in key sections: The Abstract and Conclusion do not present quantitative experimental results to demonstrate the effectiveness of the proposed method. It is suggested to include key performance metrics in these sections to strengthen this manuscript’s credibility and clearly highlight its contributions.

Author Response

Thank you very much for taking the time to review this manuscript. Please find the detailed responses below and the corresponding revisions in the re-submitted files.

Comment 1: This manuscript inconsistently uses terms such as "this study", "our work", and "our research". It is recommended to maintain consistency in terminology throughout this manuscript to improve academic clarity and readability.
Response 1: Thank you for your careful reading and helpful suggestion. We agree that consistent terminology enhances clarity. We have revised the manuscript and replaced the term 'our' with 'this study' or 'the proposed system' throughout the text to maintain consistency and improve readability.

Comment 2: Lines 139–147 introduce task skill and actor skill, but their interrelationship is not clearly explained. It is suggested to elaborate on their connection and provide a more detailed explanation of Figure 2 to enhance clarity.
Response 2: Thank you for bringing this to our attention. In our approach, the task skill generates an ideal trajectory from the agent's current state to its goal. This trajectory is purely geometric and independent of the agent's physical characteristics. On the other hand, the actor skill receives this ideal path as input and encodes the optimal motor behavior required to follow it, taking into account the current mechanical conditions of the agent. This separation allows the system to retain general task knowledge while adapting physical control in response to faults. To improve clarity, we added a summary of this relationship directly in Section 3.2 (page 5, line 190)

Comment 3: This manuscript mentions that the proposed method is applicable to safety-critical systems such as autonomous driving but does not provide specific case studies or application scenarios. It is recommended to expand the discussion in the Conclusion section to highlight the practical implications of the method.
Response 3: Thank you for the valuable comment. In particular, our method could be applied to real-world deployment contexts, such as autonomous driving systems and delivery robots, where uninterrupted operation is critical during mechanical faults. In these domains, uninterrupted operation is crucial, particularly in the event of mechanical failures. We revised the Conclusion section to emphasize the practical implications of our method. (page 15, line 424).

Comment 4: Absence of quantitative experimental evidence in key sections: The Abstract and Conclusion do not present quantitative experimental results to demonstrate the effectiveness of the proposed method. It is suggested to include key performance metrics in these sections to strengthen this manuscript's credibility and clearly highlight its contributions.
Response 4: Thank you for your constructive suggestion. In the Experiment section, we added a clarifying sentence that mentions the improvements in lateral deviation and recovery time under various failure scenarios. (page 12, line 363) Additionally, we highlight the presence of these results in the Conclusion Section to provide a more straightforward explanation of the contribution of our method. (page 15, line 417)

Reviewer 3 Report

Comments and Suggestions for Authors

Dear Author,

Thank you for your valuable research. 

The proposed methodology enables RL agents to continue functioning even under mechanical failures via disentangled dual skill variables. The concept of disentangled skill discovery and online fault accommodation differentiates the paper from existing fail-safe RL mechanisms, qualifying it as an RL breakthrough in safety-critical systems.

Nevertheless, there are some comments that need your attention.

  1. I missed a comparison with other Fail-operational Systems. The research compares its approach largely with typical reinforcement learning algorithms but does not compare directly with other fail-operational RL systems.
  2. While the approach is resilient for certain mechanical failures, the paper does not address cases with advanced sensor failures or aggressive environments that may undermine RL performance.

Recommendations:

  1. Compare with existing fail-operational RL methods to highlight relative strengths and potential weaknesses.
  2. Explore other modes of failure, including sensor failure and more complex adversarial cases, to assess robustness to a greater extent.
  3.  

Author Response

Thank you very much for taking the time to review this manuscript. Please find the detailed responses below and the corresponding revisions in the re-submitted files.

Comment 1: I missed a comparison with other Fail-operational Systems. The research compares its approach largely with typical reinforcement learning algorithms but does not compare directly with other fail-operational RL systems.

Response 1: We thank the reviewer for this insightful comment regarding the comparison with other fail-operational reinforcement learning (RL) systems. In conventional fail-operational systems, particularly in safety-critical domains such as aerospace and autonomous driving, architectures often rely on structured Fault Detection, Isolation, and Recovery (FDIR) pipelines. Notably, FDIRO [13] extends this paradigm by incorporating an optimization step after recovery to ensure the reconfigured system remains contextually effective. This strategy, adapted for autonomous vehicles, typically employs low-latency failure detection and isolation, followed by system reconfiguration to ensure safety constraints are met. The subsequent optimization phase adjusts component placement (e.g., computation node assignments) based on runtime context to enhance efficiency.

Additionally, hierarchical fallback architectures, such as those presented in [14], propose tiered control and inference strategies. These systems switch between high-performance and low-risk models depending on system confidence and environmental uncertainty. Such approaches emphasize system-level redundancy and architectural fallback mechanisms rather than the policy-level adaptability at the core of our approach.

Moreover, redundancy-based techniques, such as analytical redundancy, are commonly adopted in conventional fail-operational systems. For instance, Simonik et al.  [15] demonstrate steering angle estimation for automated driving using analytical redundancy to maintain vehicle functionality in the event of sensor failures. These methods ensure fault tolerance by using estimation models to replicate failed sensor outputs and maintain control continuity.

In contrast, our contribution introduces an innovative learning and reasoning model that implements a dual-skills approach, distinguishing between ‘task skill’ and ‘actor skill’. The ‘task skill’ component is responsible for high-level planning, such as determining the vehicle’s intended driving path or destination. In contrast, the ‘actor skill’ governs low-level control actions based on the vehicle’s current mechanical state and environment. This separation enables flexible and modular control, where high-level objectives and low-level execution are decoupled, allowing for better accommodation of system changes or faults.

This distinction positions our approach as a complementary advancement: whereas traditional fail-operational systems emphasize reactive architectural robustness, our method enables intelligent behavioral control through the modular decomposition of skills. We will clarify this comparison in the revised manuscript and expand the related work section to reflect these insights.

We revised the Introduction section to reflect the Reviewer’s comment on the comparison with other fail-operational systems. (page 2, line 40).

Comment 2: While the approach is resilient for certain mechanical failures, the paper does not address cases with advanced sensor failures or aggressive environments that may undermine RL performance.

Response 2: We thank the reviewer for bringing this vital point to our attention. We agree that while our approach demonstrates robustness to mechanical failures, such as actuator degradation or partial system faults, it does not explicitly address failure cases arising from advanced sensor degradation, sensor spoofing, or highly aggressive environments where observation quality is significantly compromised.

Our current framework assumes that the agent receives sufficiently informative state observations (even if affected by mechanical anomalies) and primarily focuses on motor-level failure or morphological changes. Extending this framework to cover perception-level failures would require the integration of techniques such as: 
·    Observation uncertainty modeling or estimation (e.g., Bayesian filters or variational inference),
·    Sensor fusion with redundancy, and
·    Robust representation learning that can generalize across input noise or occlusion.

These topics lie beyond the scope of the current work; however, we agree that they represent critical areas for future development, especially in safety-critical domains such as autonomous driving or aerial robotics. We clarified these limitations in the revised manuscript and outline future directions for incorporating sensor-level fault detection and robust adaptation in adversarial or perceptually degraded environments. (page 14, line 403)

[13] Kain, T.; Tompits, H.; Müller, J.S.; Mundhenk, P.; Wesche, M.; Decke, H. FDIRO: A general approach for a fail-operational system design. In Proceedings of the Proceedings of the 30th European Safety and Reliability Conference and 15th Probabilistic Safety Assessment and Management Conference, https://doi.org/10.3850/978-981-14-8593-0_4204-cd (Research Publishing Services, 2020), 2020.
[14] Polleti, G.; Santana, M.; Del Sant, F.S.; Fontes, E. Hierarchical Fallback Architecture for High Risk Online Machine Learning Inference. arXiv preprint arXiv:2501.17834, 2025
[15] Simonik, P.; Snasel, V.; Ojha, V.; Platoš, J.; Mrovec, T.; Klein, T.; Suganthan, P.N.; Ligori, J.J.; Gao, R.; Gruenwaldt, M. Steering Angle Estimation for Automated Driving on Approach to Analytical Redundancy for Fail-Operational Mode. Available at SSRN 4938200, 2024.